# Machine learning prediction of connectivity, biodiversity and resilience in the Coral Triangle

Lyuba Novi[1] & Annalisa Bracco [1✉]

Even optimistic climate scenarios predict catastrophic consequences for coral reef ecosystems by 2100. Understanding how reef connectivity, biodiversity and resilience are shaped by climate variability would improve chances to establish sustainable management practices. In this regard, ecoregionalization and connectivity are pivotal to designating effective marine protected areas. Here, machine learning algorithms and physical intuition are applied to sea surface temperature anomaly data over a twenty-four-year period to extract ecoregions and assess connectivity and bleaching recovery potential in the Coral Triangle and surrounding oceans. Furthermore, the impacts of the El Niño Southern Oscillation (ENSO) on biodiversity and resilience are quantified. We find that resilience is higher for reefs north of the Equator and that the extraordinary biodiversity of the Coral Triangle is dynamic in time and space, and benefits from ENSO. The large-scale exchange of genetic material is enhanced between the Indian Ocean and the Coral Triangle during La Niña years, and between the Coral Triangle and the central Pacific in neutral conditions. Through machine learning the outstanding biodiversity of the Coral Triangle, its evolution and the increase of species richness are contextualized through geological times, while offering new hope for monitoring its future.

---

[1] School of Earth and Atmospheric Sciences and Program in Ocean Science & Engineering, Georgia Institute of Technology, Atlanta, GA 30332, USA.
✉email: abracco@gatech.edu

Anthropogenic stressors, from climate change to overfishing, threaten ocean biodiversity and ecosystem functioning[1,2]. In the Coral Triangle (CT), the most diverse and biologically complex marine ecosystem, warming constitutes the greatest threat. The CT homes over 600 reef-building coral species (75% of known species); 3000 species of reef fish and 75% of known mollusks[3].

In the past 30 years, maximum and minimum ocean temperatures around the CT have risen by 0.09 and 0.12 °C per decade and will climb an additional 1–4 °C by 2100. An increase of more than 2 °C will eliminate most coral-dominated reefs. Without a sustainable pathway forward CT reef ecosystems, which in Asia make up for 25% of the yearly fish catch[4,5], may collapse by 2100[6], impacting the livelihoods of 120 million people and cutting protein supplies to one billion people.

Halting or slowing biodiversity loss in the CT requires understanding what is causing it in the first place. This problem has been long debated[7]. It has been hypothesized that the CT is a center of origin: speciation occurred within the CT from geological times, with biodiversity gradients due to the ocean currents that limit expansion outwards[8]. It was then argued that the CT owns its biodiversity to being a center of species accumulation[9], assuming that different species accumulate into the CT via dispersal through ocean currents, after speciation took place in regions outside the CT. Finally, it was theorized that the CT is a center of overlap, where the geological separation of the Indian and Pacific Oceans, allowed the formation of different fauna through vicariance, followed by Indian and Pacific sister species overlap through species-ranges expansion, thus augmenting biodiversity[10].

Here we address the biodiversity question by examining the connectivity and *ecoregionalization*[11] of the western and central Pacific and of the Indian Oceans (Fig. 1) using machine learning algorithms.

Connectivity quantifies the degree and directionality of propagules, larvae and juvenile dispersal[12,13], while ecoregionalization is useful for monitoring dispersal of pollutant or invasive species. Together they are pivotal to designating effective marine protected areas. We are interested in connectivity because large-scale larval transport and recruitment among distant reefs is key to corals' resilience, as recovery after widespread damage depends on recolonization[14–20]. We are interested in ecoregionalization because conservation management and mitigation strategies require information about the ecoregions that demark unique assemblages of species[21]. Together, they contribute to colonization and resilience of a given reef, and in turn to the biodiversity of a broader region.

In most of the world's Oceans, defining ecoregions and connectivity is complicated by data sparseness and by large-scale, time-dependent ocean currents. Accurate predictions of community susceptibility to these currents remain elusive. As circulation models and reanalysis datasets have become available and reliable, attempts have been made by simulating larval dispersal using particle tracking[22–24]. In these brute-force approaches, particles are either released everywhere searching for ecoregion boundaries, but the large spatial coverage dramatically reduces the maximum time span[22,25], or at specific reefs to quantify connectivity, but only few sites are considered[24,26]. We show that unsupervised machine learning through δ-MAPS, a complex-network algorithm developed for dimensionality reduction and network inference and related to clustering, multivariate statistics and community detection[27,28], provides a powerful alternative to infer simultaneously connectivity and ecoregionalization at the ocean mesoscale (~30–300 km) and decadal times when applied to sea surface temperature anomalies (SSTa)[29]. When further augmented by community detection and PageRank centrality, δ-MAPS networks inform about the resilience potential of the ecoregions, defining a novel framework to identify where and when mitigation strategies will be most beneficial to the survival of reefs in the CT (Fig. 2).

## Results

The intuition of leveraging SSTa for ecoregionalization and connectivity purposes using δ-MAPS exploits the dynamical relationship existing between SSTa and sea surface height anomalies (SSHa), and therefore currents, at spatiotemporal frequencies pertinent to the transitions of marine communities at latitudes explored in this work (see Materials and Methods). A verification of the SSHa-SSTa high correlation, which provides the desired link of SSTa to the surface ocean advective properties is presented in Supplementary Fig. 1. In our work, the autonomous and unsupervised identification of spatially-contiguous regions (i.e. "domains") characterized by a highly correlated temporal activity in SSTa and their mutual physical connectivity is carried out through δ-MAPS. The resulting weighted functional network, connecting any two regions, quantifies the role of a given domain in the large-scale connectivity (see Materials and Methods for details on δ-MAPS and network metrics).

We build upon this approach, applied to the Mediterranean Sea by the authors in[29], accounting for the specificity of the CT climate: ocean temperatures, currents and therefore connectivity patterns in the Indo-Pacific basin are strongly influenced by the El Niño Southern Oscillation (ENSO). ENSO, in its warm and cold phases, El Niño and La Niña, not only drives the most dramatic year-to-year variation of the Earth's climate system but also is responsible for extensive coral bleaching and up to 97% coral mortality[30]. In the following, we search for ecoregions using δ–MAPS applied to SSTa from the GLORYS12V1 1/12° CMEMS global ocean eddy-resolving reanalysis product[31,32] over the period 1993–2017 (see Materials and Methods). Years are defined from April 1st to March 31st, to separate among ENSO phases, with 8 years in each category of neutral, El Niño and La Niña events (Supplementary Fig. 2 and Supplementary Table 1).

**Ecoregions and the El Niño Southern Oscillation**. The ecoregions inferred with δ-MAPS are reported in Fig. 3 for a maximum competency period $\tau_{max}$ of two months, but results are confirmed if one or three months are assumed instead (see Supplementary Figs. 3 and 4). The ecoregionalization is consistent with previous boundaries based on coral distributions[3,24], while

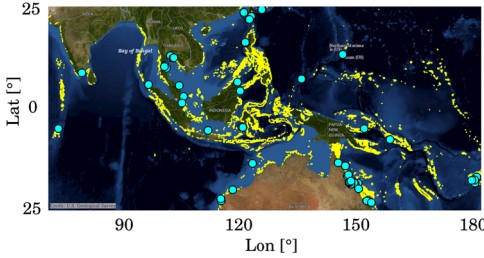

**Fig. 1 The CT and Indian Ocean study area, with superposed coral reef and spawning locations.** In yellow the distribution of coral reefs in the study area, as obtained from the UNEP-WCMC (UNEP-World Conservation Monitoring Centre), WorldFish Centre, WRI (World Resources Institute), TNC (The Nature Conservancy) Global distribution of Coral Reefs database[66–69]; in cyan known coral spawning locations within the study area, from the Coral Spawning Database[65]. Topographic and bathymetric background: USGS Imagery Topo, courtesy of the U.S. Geological Survey.

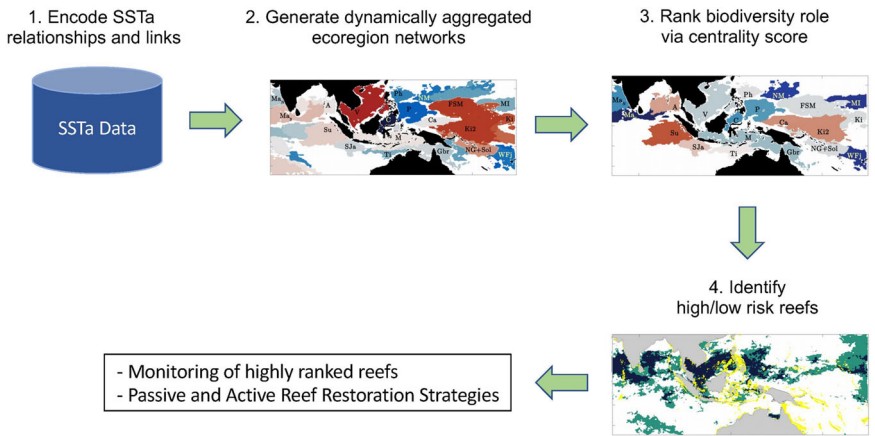

**Fig. 2 Conceptual framework and workflow.** Schematic representation of the proposed ecoregionalization, connectivity and resilience framework.

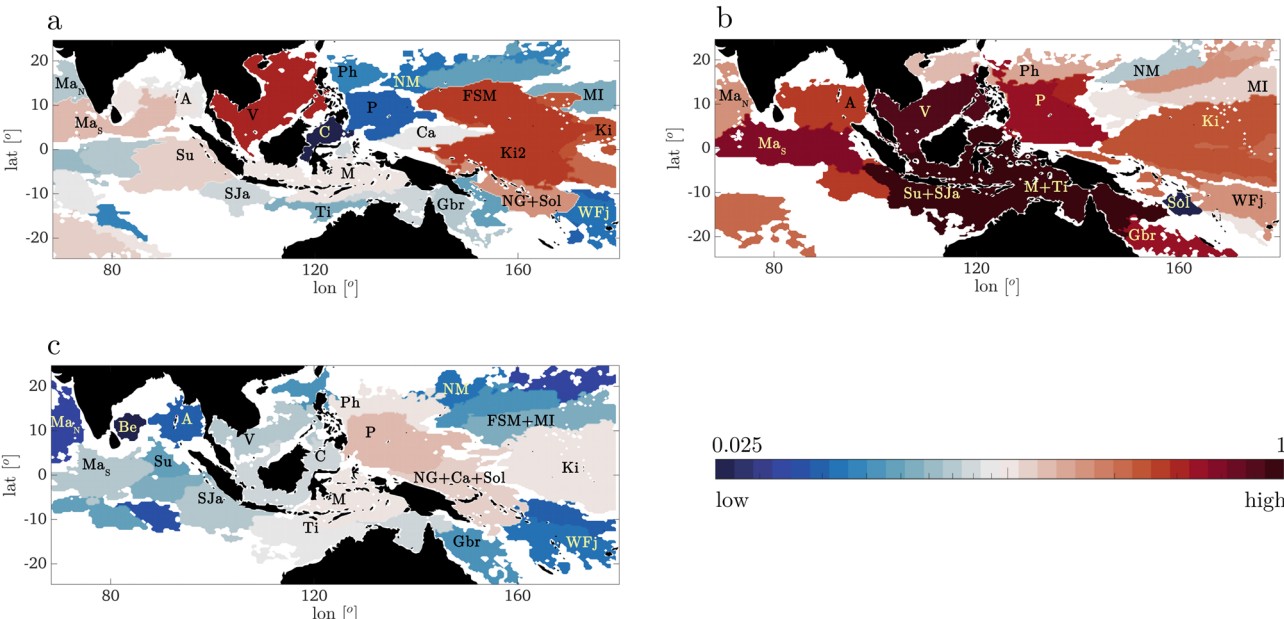

**Fig. 3 Inferred domains.** Domains over the period 1993–2017 in the CT and Indian Ocean for $\tau_{max} = 2$ months for aggregated (**a**) El Niño, (**b**) La Niña, (**c**) neutral years, colored according to their strength value. In each period, domains with strength below the 20th percentile (computed for that period) are not shown. Strengths are normalized by $6 \times 10^6$, and the color scale is logarithmic.

extending further both in space and time any previous study for this area.

ENSO modulates the distribution of δ-MAPS domains and their strength, leading to three distinct patterns. Briefly, in neutral years domains along the Equator have similar strengths; during El Niño years several domains around the distant Pacific islands (FSM, Ki, Ki2) strengthen, indicating increased connectivity; during La Niña years most CT ecoregions are stronger, wider, and often overlapping. In both ENSO positive and negative phases, the ecoregion between northern Papua New Guinea and the Caroline Island weakens compared to neutral conditions, in agreement with previous outcomes from Treml and colleagues[33]. El Niño and La Niña conditions modify in opposite ways the ecoregions surrounding the Makassar Strait that regulates water exchanges between the western Pacific and eastern Indian Ocean[34], with La Niña causing an increase in strength and size of the ecoregions as well as interbasin connectivity. In the Indian Ocean, under La Niña conditions the domain surrounding the southern Maldives Islands (Ma$_S$) gets wider and stronger and extends eastward reaching as far as the southern border of Andaman Islands ecoregion (domain A).

Having established the ecoregionalization of the CT and Indian Ocean, we further group domains sharing strong connections into communities, in which we expect exchange of genetic material to occur at least for a given ENSO phase by means of a community detection algorithm that exploits a centrality-base clustering[35]. These *supercommunities* are shown in Supplementary Fig. 5. In neutral conditions ecoregions are distributed nearly uniformly among 4 communities, broadly covering the Indian Ocean (C1), the main Coral Triangle (C2), the eastmost Pacific islands in the area considered (C3), and the southern and western Pacific (C4). In El Niño years the communities continue to be 4, but their interconnectivity weakens, while in La Niña years only two unrelated supercommunities are identified, with the largest including most of the CT, the Indian Ocean and several of the west Pacific islands. Links among supercommunities are especially important, as their removal would imply isolation among extended areas. However, in the CT and Indian Ocean the ENSO variability ensures that no single link is indispensable for genetic exchanges across the domain considered, because ecoregions, the communities and their connectivity all change greatly under different ENSO phases.

Given these dynamical conditions, we revisit the concept of Island Biogeography, first introduced by the work of MacArthur and colleagues[36]. This idea relates the number of species present on an island to a balance between immigration rate of new species and extinction rate, noting that the number of species at equilibrium generally increases with the island's size and proximity to a source of immigrant species. Our supercommunities can be thought of as dynamical islands whose shape and position change over time due to ENSO variability. The four "islands" found in neutral and El Nino years merge into two larger ones during La Nina years, exchanging colonizers periodically. Supercommunities at different $\tau_{max}$ are shown in Supplementary Fig. 6.

**Biodiversity patterns: ENSO as driver of the CT biodiversity.** What does make an ecoregion biodiverse? From a connectivity perspective, we hypothesize that the ecoregion should be a point of arrival for a lot of connections. Larvae can be transported to this area from many ecoregions, even if each of these connections is not from a biodiverse area, or from few biodiverse regions. This hypothesis is directly linked to the potential maintenance of high biodiversity in the CT.

Identifying areas which are arrival for a lot of connections is a problem parallel to assessing the popularity level of web sites and quantifying the likelihood to reach any given web page by randomly following links on the internet and can be solved using the PageRank Centrality[37] algorithm. The PageRank Centrality quantifies if a webpage (or a domain in our case) is reached by a high number of connections with a low centrality value each, or one/few connections from pages (other domains) with high centrality, or both. We therefore apply it, using the implementation provided in the Wolfram Mathematica software[38], to the positive unweighted network links identified by δ-MAPS among the labeled domains of Fig. 3 whenever the correlation coefficient is ≥ 0.35.

For each ENSO phase, centralities are computed as solutions to $c = \alpha_F \ a^T \cdot d \cdot c + \beta$ (see[38]), where $a^T$ is the transpose of the adjacency matrix of the selected network, $d$ is a diagonal matrix with elements $1/\max(1, d_{i\text{-}out})$, $d_{i\text{-}out}$ being the out-degree of the $i^{th}$ δ-MAPS domain of that network, $\alpha_F$ is a damping factor (here

0.85, but robustness has been verified for $\alpha_F$ from 0.5 to 0.9 in Supplementary Fig. 7), and β contains the initial centralities (here always equal to $1/N_{nodes}$, with $N_{nodes}$ being the number of domains -or nodes- in the selected network). We also computed an averaged *biodiversity score* by selecting the ENSO phase with the most domains (El Niño years) and computing, for each domain, the average of all three centralities in that geographical area. The resulting vector represents the mean contribution of connectivity to the maintenance of the overall coral biodiversity, in each location, computed accounting for ENSO-phase aggregations over 1993–2017. The higher the biodiversity score, the higher is the connectivity-modulated potential for biodiversity maintenance. The biodiversity score in each ENSO phase is reported in Fig. 4, while the average scores can be found in Supplementary Fig. 8. The highest values are found over the Coral Triangle and gradually decrease for increasing distance from it, consistent with previous works (see for example Fig. 3 in the work of Veron and colleagues[3]). Our scores, based on the direction of the physical transport, highlight that the CT is a point of arrival of genetic diversity, and La Nina years are more conducive to biodiversity than all other times. The evolutionary origin of the CT remains under debate, but today ENSO variability plays a pivotal role by promoting different connectivity pathways into the CT depending on its phase, and by promoting ecoregions expansion and overlap in its negative phase. When focusing on coral species biodiversity, our analysis supports the *center of accumulation* and *center of overlap* hypotheses under current conditions with ENSO being a critical contributor. These findings are robust and verified also for $\tau_{max} = 1$ and 3 (Supplementary Fig. 9).

**Connectivity-modulated bleaching resilience.** Previous works on coral resilience and recovery capacity point to the importance of external recruit supply for rapid recovery of reefs. Here, we propose a metric to estimate the recovery potential that accounts for the combined effect of connectivity and time cumulative bleaching stress. We do so by merging the newly obtained connectivity information with composites of the Bleaching Alert Area (7-day maximum) (*baa*) product from the NOAA Coral Reef Watch's (CRW) Version 3.1. (ref. [39], while the works of Liu and

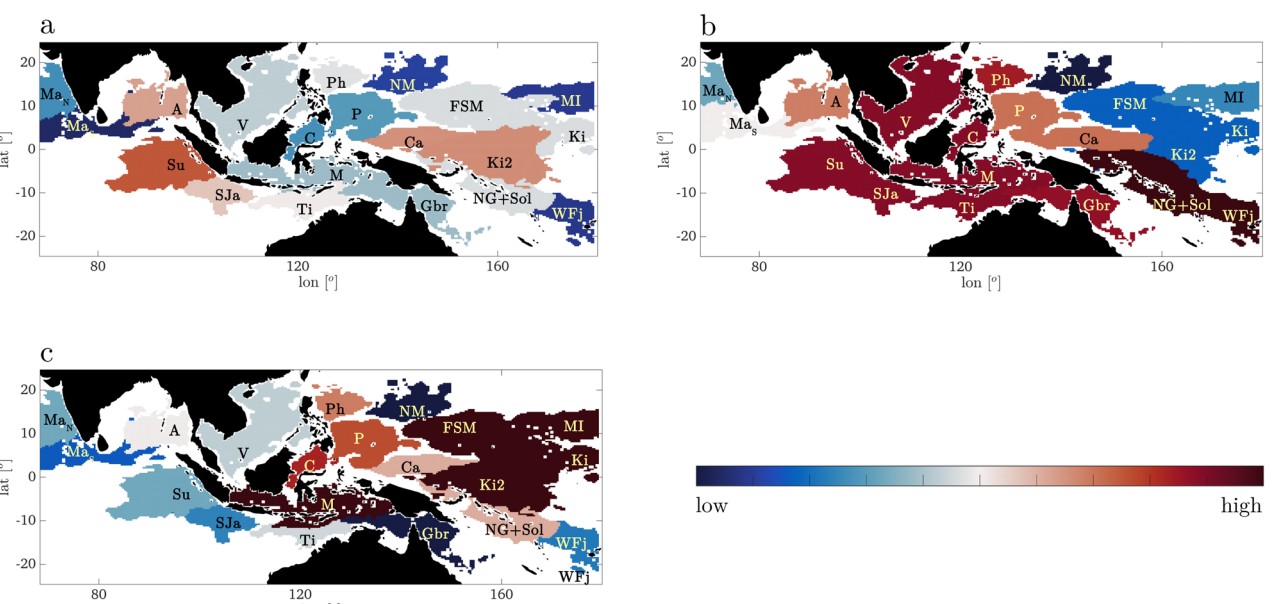

**Fig. 4 Estimated Biodiversity Score over 1993–2017.** Domains found during El Nino years, colored according to their Biodiversity Score in (**a**) El Niño years, (**b**) La Niña years, (**c**) neutral years. The color scale is linear (low = 0.03, high = 0.08).

colleagues[40,41] provides an overview of the product). First, we compute a time-cumulative *baa* index (*t.s.baa*) for coral bleaching stress (see Methods for its calculation in the various ENSO phases). The *t.s.baa* is amplified in El Niño and La Niña years compared to neutral conditions (Supplementary Fig. 10), in agreement with previous works[42,43], with the South China Sea and the coasts of Sri Lanka being the exception in this amplification. Second, within each identified domain, we define a recovery probability following a bleaching event that depends on the cumulative effect of bleaching or number of bleaching occurrences (i.e. *t.s.baa*) and its connectivity potential. As a result, a domain hosts a reef capable of rapid recovery if the time cumulative bleaching pressure is low (at any connectivity level), or the potential connectivity is high despite a high *t.s.baa*, or high potential connectivity and low time cumulative bleaching pressure (low *t.s.baa*) act in concert.

We express the relative importance of *t.s.baa* and the potential connectivity as a *connectivity modulated bleaching vulnerability* or $CMBV = t.s.baa / \sigma^+$, where $\sigma^+$ is the domain strength computed with only incoming and undirected positive links. For domains that overlap we consider only the highest value of *CMBV* at the overlapping grid points. Low (high) values of *CMBV* correspond to high (low) chances to find a connectivity resilient reef. For each ENSO phase, a *candidate score of 1* is assigned to the grid points with *t.s.baa* and/or *CMBV* below the 25th percentile of possible values (low cumulative bleaching, or high connectivity modulated bleaching resilience), with all thresholds computed in neutral years, and at the highest resolution for *t.s.baa*, and a *score of 0* is given elsewhere. Lastly, we sum the three matrices, one per each ENSO phase, obtaining in each grid point a *recovery potential score (RPS)*. Values of $RPS = 3$ identify areas with a high probability to find a reef that may rapidly recover after a disruptive event, because the combination of *CMBV* and *t.s.baa* is favorable, regardless of the ENSO phase. Areas where $RPS = 2$ may also be considered targets for rapid recovery but to a lesser extent, because of a detrimental combination of cumulative bleaching stress and potential connectivity in one ENSO phase. Figure 5 shows the *RPS* map for areas with high recovery potential with coral reef locations superposed (see Supplementary Fig. 11 for $\tau_{max} = 1, 3$). The *RPS* metric contextualizes the observations of bleaching-resilient reefs targeted as conservation priorities by Darling and colleagues[44] and by the "50-Reef Project"[45], including the "Super-Reefs" of Racha Noi and Rock Island (Palau). For about 70% of the sites identified in the cited works, physical connectivity is crucial in modulating bleaching resilience. Indeed, an equivalent index based only on temperature (= low t.s.baa) would not capture most of them (Supplementary Fig. 12). Nearly identical results are found if the NOAA Coral Reef Watch (CRW) monthly composites of maximum Degree Heating Week (DHW) are used instead of the *baa* product as estimate of thermal stress of corals. DHW is based on satellite-derived SSTs and in-situ temperature loggings and is available daily at 5 km resolution. The DHW monthly composites are processed as follows: First, we associate numerical values to three classes of equivalent potential bleaching risk: for $0 < DHW < 4$, with 0 indicating "no stress" and 4 "warning" for potential bleaching risk, we assign a value of 2; for $\leq 4$ $DHW < 8$, when consequential bleaching is expected, we assign a value of 3; for $DHW \geq 8$, or widespread bleaching and likely mortality, we assign a value of 4; we assign 0 elsewhere. Second, in each ENSO phase, we compute a time cumulative value of these DHW classes (*t.s.dhw*), and third we evaluate the CMBV as the ratio between *t.s.dhw* and $\sigma^+$. Finally, the RPS computation is carried out as before, and shown in Supplementary Fig. 13.

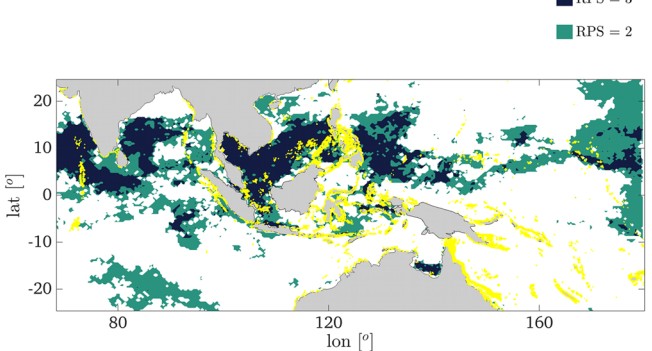

**Fig. 5 Recovery potential score (*RPS*) over 1993–2017.** Dark blue (green) areas identify domains with RPS = 3 (RPS = 2), while known reefs are in yellow. In the Indian Ocean, the northern Maldives Islands, the Laccadive archipelago, and the reefs around Sri Lanka, together with smaller areas to the south-west of Thailand, the southern Nicobare Islands, and western Sumatra have a high RPS. In the South China Sea, recovery is likely along the coasts of southern Vietnam, west Cambodia, and southern Thailand, and around Riau Islands and northern Spratly Islands. A portion of reefs in the Indonesian Bangka-Belitung province has also a high RPS, along with reefs in central Java. In the northern CT, only reefs in the Sulu Sea have a high *RPS*, while in the central CT the Maluku Islands and Western Papua, Nusa Tenggara (eastern Indonesia) and south-eastern Sulawesi show good recovery potential. Finally, high *RPS* values are found to the south-east of the Philippines, in a small portion of northern Australia, and in some Pacific islands such as Palau, the western Federate States of Micronesia, and to a lesser extent the southern Marshall Islands and Tuvalu.

## Discussion

Coral reef ecosystems have changed profoundly over time and have evolved and adapted to environmental and anthropogenic stressors across millennia. Today, however, they are threatened with rapid extinction due to fast warming ocean temperatures. Their survival requires establishing appropriate monitoring and sustainable management practices, and doing so requires understanding how reef connectivity, biodiversity and resilience are shaped by climate variability.

Efforts to define the ecoregions in the Coral Triangle (CT), the most biodiverse ocean hotspot, have been traditionally based on taxonomic and ecological approaches, but direct observations of species distributions are sparse in space and time, concealing the relationships between ecoregions, climate variability and ocean circulation over the basin-wide and decadal spatio-temporal scales needed to understand evolution and resilience potential. Direct numerical simulations of ocean currents and larval transport suffer from similar limitations: if the resolution is enough to resolve the relevant dynamics, their time span is too short to explore interannual variability. This work proposes a framework based on machine learning to overcome these problems and evaluate the concurrent role of physical connectivity and ENSO—the dominant climate mode in the region—in shaping the biodiversity of the CT and surrounding areas. It is based on sea surface temperature anomalies and unsupervised network analysis, and allows for assessing both connectivity and bleaching recovery potential of reef ecosystems over unprecedented large space and time scales with a uniform coverage.

Gaining a dynamical view of coral connectivity, and quantifying how spatial and temporal variability impacts connectivity, open unprecedented opportunities to improve sustainable management practices for biodiversity conservation[21,29]. This knowledge is especially important when biodiversity must rebuild following a devastating damage, as is occurring even more often in the CT.

Furthermore, our results suggest that the extraordinary biodiversity observed today results from the CT being both a center of accumulation and a center of overlap, with ENSO variability positively and strongly contributing to the observed biodiversity patterns. While ENSO has long been associated with coral reefs mortality, due to the prolonged higher-than-normal temperatures experienced in some areas, our work reveals that it is highly beneficial to biodiversity, enhancing the large-scale exchange of genetic material between the Indian Ocean and the CT during La Niña years, and between the CT and the area to the east in neutral conditions.

Investigating the biodiversity origin conundrum, Renema and colleagues[46] introduced the concept of "hopping hotspots" that accounts for the influence of plate tectonics on the origin, evolution and vanishing of marine biodiversity hotspots. They showed that the "relocation" of biodiversity hotspots is concurrent with paleo-geography, pointing to the formation of an Indo-Pacific hotspot after the early Miocene, and the appearance of the first fossils records for coral genera at the Miocene-Pliocene transition. The biodiversity evolution was concurrent not only with the restriction of the Indonesian Throughflow, which assumed a geological setting closer to present days, but also with paleorecords of ENSO-like variability along the Equator. Indeed, ENSO-like events have been documented during the Pliocene and Pleistocene and as early as at the Miocene-Pliocene transition[47], with coral proxy data disproving the hypothesis of a permanent El-Nino condition during the Pliocene[48]. Model simulations also support ENSO-like variability in the Miocene with much longer periodicity and greater amplitude than today, and a warming pool extending well into the Indian Ocean[49]. Our findings indicates that geological changes may have imprinted the Indo-Pacific biodiversity through the associated climate variability signal, with ENSO-like variability promoting the inflow of genetic diversity by contributing both larval supply and ecoregions overlap in the CT. The increase in ocean biodiversity through geological times pointed in[46] would be supported by this interpretation, and so would be the evolution of species richness in the CT from the last glacial maximum to pre-industrial times[50], given the consistent, gradual intensification of ENSO amplitude throughout the Holocene[51] in response to changes in insolation. A consequential result is that future change in ENSO amplitude, frequency, or pattern may modify these dynamical linkages, and should be accounted for in light of their connectivity implications, not just for their climate influence.

This work establishes a framework for evaluating feasibility and relevance of targeted conservation efforts in the CT and surrounding areas by establishing the causative process(es) responsible for the CT biodiversity, evaluating the recovery potential of reefs in the CT and surrounding areas, and identifying the reefs that are central to its overall biodiversity resilience.

The proposed approach brings about an effective and fast tool to explore a rather complex set of processes acting in concert but, by its nature, introduces important simplifications to the biological-physical interactions at play. For example, we neglect the species-specific nature of larval mortality and settlement competency characteristic[52] and use only information about pelagic larval duration (PLD) (see Methods). The paucity of detailed data over the whole study area justifies the usage of the PDL alone, but for single-species studies will not be sufficient. Additionally, the proposed analysis of connectivity modulated recovery potential through CMBV and RPS, while effectively extending the current methodology for reefs prioritization, necessarily simplifies coral response and adaptation to thermal stress over time. In this work, bleaching recovery is elevated (at any connectivity level) if the time of cumulative exposure to previous bleaching is low, as low exposure implies higher coral cover, thus a more stable community composition in a given ecoregion, justifying a higher supply of larvae. However, we do not account for the "stress-hardening" effect, according to which corals acquire and maintain stress tolerance through past environmental stress[53–58], making low-exposed communities potentially more vulnerable to future bleaching. Nonetheless, the proposed framework helps elucidating the first-order mechanisms underlying the existing large-scale connectivity-modulated bleaching resilience, contextualizing many observations of resilient reefs.

By merging physical intuition and machine learning, we provide an integrated and more effective approach to managing coral networks, promoting larval replenishment and monitoring ecosystem stability[59]. The ecoregion identification and biodiversity score, taken together with the recovery potential metric, provide indeed a powerful tool to identify appropriate temporal and spatial windows where connectivity-based restoration efforts and monitoring should be prioritized. Recovery potential evaluations under future bleaching scenarios may benefit from using numerical models for the computation of SSTa ranges, ENSO behavior and bleaching warnings values, rather than projecting the current conditions into the future.

## Materials and methods

**δ-MAPS: ecoregions and connectivity inference through SSTa.** The physical reasoning that allows to use SSTa to address ecoregionalization and large-scale connectivity with a complex network tool like δ-MAPS is rooted in the following consideration: At equatorial, tropical and mid-latitudes SSTa are characterized by strong coupling with the layers underlying the ocean surface through horizontal oceanic currents. Indeed, dynamical links relate the temporal variability of SSTa to that of sea surface height (SSH) anomalies and therefore currents[60], with the SST observations being available at higher resolution than SSH fields, especially in coastal areas. The use of SSTa in turn facilitates exploring the connectivity question at time and spatial scales covered in this work, because they directly relate to the mesoscale (30–300 km) flow advective properties. δ-MAPS is a machine-learning complex network algorithm for dimensionality reduction, which allows to identify in an autonomous and unsupervised manner spatially contiguous regions or domains and their mutual connectivity, while accounting for autocorrelations. Given a spatio-temporal field $X$(t), δ-MAPS identifies spatially contiguous domains that share the same dynamic functions, i.e. share a highly-correlated temporal activity, and then unravels the connectivity linkages among them. A functional network is constructed based on these connections, weighting each edge to reflect the magnitude of interaction between any two domains. The strength of a domain is finally defined as the sum of all the absolute weights of all the edges pertaining to that domain. δ-MAPS is well suited to identify ecoregions if applied to a field that contains information of the flow dynamics. With relevance to this project, δ-MAPS identifies the structural connection among SSTa domains. Links define the physical connectivity between any two domains and the sum of the absolute weights of all edges of a given domain quantifies its role in the large-scale connectivity.

δ-MAPS works through two steps: domain identification and network inference[27,28]. Domains are hypothesized to have *epicenters* or *cores*, where their local homogeneity is highest. Cores are identified on a gridded dataset requiring that the local homogeneity of a grid cell is a local maximum and greater than a threshold δ. Cores are then expanded and merged to identify domains. For each domain $A$, its signal $X_A(t)$ is then defined as the weighted cumulative anomaly of all the time series within A:

$$X_A(t) = \sum_{i=1}^{|A|} x_i(t) \cos\phi_i \qquad (1)$$

where $X_i(t)$ is a time series of length $T$ associated to grid cell $i$ with latitude $\phi_i$ and $|A|$ is the number of grid cells in A. To infer the network of a given set of domains, we compute the Pearson correlation $r_{A,B}(\tau)$ between each possible pair of domains A and B for a lag range $\tau \in [-\tau_{max}, \tau_{max}]$. Each pairwise correlation is tested for a given significance level after accounting for autocorrelations using the Bartlett's formula[61]. Two domains A and B are connected if there exists at least one significant correlation between the two at any lag in the range $\tau \in [-\tau_{max}, \tau_{max}]$, denoted as $R_{A,B}(\tau)$. The link is assumed undirected if $R_{A,B}(\tau)$ includes the lag $\tau = 0$, whereas the connection is directed from $A$ to $B$ ($B$ to $A$) if $R_{A,B}(\tau)$ is strictly positive (negative). We assign a weight $w_{A,B}$ to each link based on the covariance between the two signals $X_A(t)$ and $X_B(t)$ at the lag $\tau*$ at which their significant correlation $r_{A,B}(\tau)$ is maximized. Finally, a non-dimensional strength value is defined for each domain as the sum of the absolute weights of all the connections incident to that domain.

In this study, δ-MAPS is applied to detrended monthly sea surface temperature (SST) anomalies over 1993-2017 from the GLORYS12V1 CMEMS global ocean

eddy-resolving reanalysis product[31,32], available at horizontal resolution 1/12°, here remapped at 1/3° without losing information given the scale of the mesoscale currents at this latitude (Supplementary Fig. 14). GLORYS12V1 covers the altimetry period (from 1993 onward) and assimilates by means of a reduced-order Kalman filter satellite observations of SST, sea level anomaly, sea-ice concentration, and in-situ salinity and temperature profiles. SST are detrended over 1993-2017 and aggregated in neutral, El Niño and La Niña ENSO years (defined from April 1st to March 31st), so that each ENSO phase is composed by 8 aggregated years. The seasonal cycle is removed in each aggregation separately (see[62] for a justification). El Niño, La Niña and neutral years are selected based on the ONI index calculated over the region (5°N-5°S, 170°W-120°W)[63,64] applied to the GLORYS SST data, requiring that the 3-months running mean SST anomalies exceed $+0.5$ °C or $-0.5$ °C for at least five consecutive months for El Niño or La Niña occurrences, respectively (Supplementary Fig. 2). In δ-MAPS, the significance level for the network inference is set to 0.03, tested using a $t$-test, and a $K$-neighborhood of 8 grid cells. The δ threshold is inferred using a significance level $\alpha = 10^{-3}$[27,28]. We consider a maximum pelagic larval duration (PLD) of 60 days by setting set $\tau_{max} = 2$ months, but the connectivity networks and our results are verified for $\tau_{max} = 1$ or 3 months.

**Coral spawning data processing**. The timing of corals spawning in the CT and central Indian Ocean controls larval availability. The geographical extension of the study area (68.33°E–180°E, 24.67°S–24.67°N) and the variety of coral species in this region preclude us from assuming a common spawning period for the entire area. Therefore, we analyzed a collection of multiple spawning data, available through the "Coral Spawning Database" (CSD)[65] from multiple sources over the entire Indo-Pacific. The CSD collates 6178 observations of coral spawning dates, for more than 300 scleractinian species from 101 locations. The CSD data ("SpawningObservationBySite" entry) are processed as follows: first, only spawning events occurred within our study area are selected; second, spawning events are aggregated by month of spawning over 1993–2017; third, the aggregation is repeated separately for neutral, El Niño, and La Niña years, obtaining three distributions (Supplementary Fig. 14 and Results). Each ENSO-phase distribution is then compared with the total one, and in each histogram, months with the number of spawning events greater than zero but below 10% of the maximum number of spawning occurrences (of that histogram) are categorized as "minor spawning", and months with a number of spawning events above that threshold as "major spawning". When considering the entire region of interest, no month can be excluded in the connectivity computation for all the ENSO phases if the PLD is 30 days or longer (Supplementary Fig. 15).

**Coral bleaching stress data processing**. Composites of monthly maximum bleaching alert area (baa) from the NOAA Coral Reef Watch's (CRW) Version 3.1 heat stress monitoring products[39] at daily frequency and 0.05° degree resolution are used to evaluate spatial patterns of coral bleaching threat over neutral, El Niño and La Niña years between 1993 and 2017. The baa composites outline the portions of the global ocean where coral bleaching heat stress hits levels of increasing severity, from "No stress" and "Watch", both set to zero, to "Warning", "Alert level 1", and "Alert level 2", set to 2, 3 and 4 respectively (see https://coralreefwatch.noaa.gov/product/5km/methodology.php#baa for details). Maps of time cumulative baa (t.s.baa) are found by summing over time the values at each pixel and are bilinearly interpolated over the SST grid.

**Reporting summary**. Further information on research design is available in the Nature Portfolio Reporting Summary linked to this article.

## Data availability

The analyzed data (SST and SSH) are obtained from the Global Ocean Physics Reanalysis GLORYS12V1 1/12° product[30] MERCATOR GLORYS12V1 (global-reanalysis-001-030-monthly), available with free access through the E.U. Copernicus Marine Service Information (CMEMS) portal[31] (https://doi.org/10.48670/moi-00021). The coral spawning data are from the "Coral Spawning Database"[65] available at https://doi.org/10.25405/data.ncl.13082333.v1. Composites of monthly maximum bleaching alert area are from the NOAA Coral Reef Watch's (CRW) Version 3.1 heat stress monitoring products[39] at daily frequency and 0.05° degree resolution, available at https://coralreefwatch.noaa.gov/product/5km/. The MATLAB codes for all analysis and to recreate the graphs in the paper and in the Supplementary Material are publicly available through figshare (https://doi.org/10.6084/m9.figshare.21587199.v1).

## Code availability

The δ-MAPS software (java version) is available at https://zenodo.org/record/7320416#.Y3LesoLMLdo, https://doi.org/10.5281/zenodo.7320416.

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

## Acknowledgements
The authors are grateful to Mark Hay for useful comments, to Fabrizio Falasca for his contributions to δ-MAPS, and to two anonymous reviewers for their insightful comments. AB was supported through a Faculty Development Grant by the Georgia Institute of Technology. LB is supported by the Department of Energy, Regional and Global Model Analysis (RGMA) Program, Grant N.: 0000253789. Conceptualization benefitted from the KITP Program "Machine Learning and the Physics of Climate" supported by the National Science Foundation under Grant No. NSF PHY-1748958.

## Author contributions
Conceptualization: L.N., A.B. Methodology: L.N., A.B. Investigation: L.N., A.B. Visualization: L.N. Supervision: A.B. Writing—original draft: L.N., A.B. Writing—review & editing: A.B., L.N.

## Competing interests
The authors declare no competing interests.
