## [Peer Review File · Communications Biology]

Reviewers' comments:

Reviewer #1 (Remarks to the Author):

This paper proposes a framework based on machine learning to overcome these problems and evaluate the concurrent role of physical connectivity and ENSO in shaping the biodiversity of the CT and surrounding areas. Their analysis is based on sea surface temperature anomalies and unsupervised network analysis and allows for assessing both connectivity and bleaching recovery potential of reef ecosystems over unprecedented large space and time scales with uniform coverage. I think this work is interesting, innovative, and well-written. Overall, I have no major concerns about this paper. Yet there are some minor issues that need to be addressed before being accepted.

Minor comments:

Line 69: 'δ-MAPS', what does this abbreviation refer to? This term was not defined earlier.

Fig.1: Both the units for latitude and longitude are missing.

Line 79: I don't think everybody knows what 'UNEP-WCMC', 'WRI', 'TNC', 'USGS' represent. Full names are needed.

Line 92-114: The first two paragraphs in the results section were more about the methods rather than the results. I would suggest moving much text to the methods while keeping a brief description of the main approach used to generate the results.

Line 132-133: 'in agreement with(32)'

Line 163-164: 'first introduced by(35)'

Line 182: 'implementation in(37)'

Line 197: 'see for example Fig. 3 in(3)'

Line 218: '38, and 39,40 for an overview of the product'

Line 244: 'as conservation priorities by(43)'

Such a way of citation seems a bit strange. When I read these sentences, they look unfinished, as the numbers do not really belong to those sentences, but rather link to the references.

Reviewer #2 (Remarks to the Author):

The manuscript brings an interest framework to assess connectivity at large scale and derive a spatial-prioritisation mechanism with potential to inform decision-making. The study derives from previously published work by the authors and looks at Sea Surface Temperature Anomalies (SSTa) over time and space to identify spatial domains or regions that likely to be highly correlated temporal activity. Based on derived correlations between SSTA and currents, the authors used 60 days as the Pelagic Larvae Duration, or time the coral larvae spends in the water column, to identify the connectivity among reefs and then perform the clustering of regions across the time series, discretised by ENSO phases. Using the inferred domains or regions, the authors expanded the work in two ways. First, by exploring the hypothesis that ENSO phases drive biodiversity in the Coral Triangle by exploring the contribution of connectivity using a centrality analysis. Second, the authors infer the recovery of reefs to bleaching events based on the exposure to cumulative thermal stress predictions (Bleaching Alert from NOAA) and connectivity levels.

Overall, the manuscript is well-written, presented in a logical format to facilitate the interpretation despite the complexity of the topic and the approach is well-supported by previous research. This work presents a valuable contribution in terms of spatial and temporal variability in the definition of bioregions and connectedness across reef systems. Well done!

However, the following points need to be considered and addressed in this work before it follows its

natural course to publication:

- Assumptions and weaknesses in the approach need to be clearly stated and discussed. While the study is supported by body of research, this work is highly correlative in nature and make use of several assumptions that should be clearly discussed to outline further work and potential weaknesses. For example, I would appreciate a discussion considering, but not limited to, the following points:

- o The effect of non-anomalous years, in terms of SST, on the patterns of connectivity. The study bases the regional clustering on SST anomalies. Would there be any difference in the spatial patterns if considering connectivity during periods when SST is within its climatological range?

- o The effect of larval mortality and settlement competency characteristics in defining population connectivity. Previous studies suggest marine population connectivity is primarily influenced by the duration of the pelagic larval stage (PLD, considered here), but also the time-bound mortality of larvae and the extent of the pre-competency window. I must acknowledge that such information is species specific, and the availability of such data is sparse and a simpler approach, such as the one used here is justified. However, it is important to acknowledge the limitations of this simpler approaches.

- o See points below about biodiversity and recovery potential.

- Centrality of networks do not equate biodiversity. While the assessment of centrality is strongly linked to the potential of maintaining biodiversity, this works appears to use centrality as an estimate of biodiversity. I loved the approach of using web-page network analyses to explore the contribution of connectivity across ENSO phases to identify the contribution from different bioregions. The discussion is very appropriate from the potential of view of temporal variability in the contribution of larvae across different bioregions to potential generate high biodiversity over geological periods, but the point of this analyses should be centred on the maintenance of such biodiversity. It is possible that I misunderstood this point, but if so, it is a point that needs clarification. Also, the authors refer to two important hypotheses, center of accumulation and center of overlap, but does not expand on them. This point needs more clarity.

- Recovery potential based on exposure to bleaching warnings. It is well-agreed in the literature that reefs that are highly connected are more likely to recover, and the authors make a good case of utilising this point to expand on this point to explore a mechanism to prioritise reefs based on their recovery potential. However, the authors also use Bleaching Alert Area datasets to explore time cumulative bleaching stress and argue that the recovery to bleaching depends on the previous number of bleaching occurrences, as well as connectivity. To this point, the authors suggest that low levels of previous bleaching events is an indicative of rapid recovery, while also considering connectivity. While there is strong support for the latter point (i.e., connectivity support recovery), I have concerns about the use of Bleaching Alert Area and the use of bleaching exposure to explore recovery potential for the following reasons:

- o What justifies the assumption that low cumulative exposure to previous bleaching events equates to rapid recovery to a bleaching event? A body of work in the literature has documented the effect of 'stress hardening', suggesting various mechanisms where coral acquire and maintain stress tolerance through past environmental stress (see references below). Perhaps, it could be argued that low exposure will mean that coral cover is higher and community composition is more stable in the ecoregion, therefore justifying a higher supply of larvae that augments to the point around connectivity. However, based on the statement around stress-hardening, these communities could also be more vulnerable, or less resistant, to future bleaching events, which could affect the rate of recovery.

- o If previous exposure to bleaching is justified, why using the Bleaching Alert Area instead of actual metrics of thermal stress (e.g., SSTA and DHW)? Bleaching Alert Area from NOAA Coral Watch is a forecasting tool to based on Degree Heating Week thresholds over a moving window of 7 days to provide a warning system for potential bleaching events. Alert 1 and 2 values are based on SST levels above documented thresholds for thermal stress and probability of bleaching and mortality (DHW >4 and >8, respectively). However, for hindcasting historical bleaching events, DHW and SSTA would be more appropriate.

o Perhaps the previous analysis on network centrality would be more appropriate here as a proxy for maintenance of genetic diversity across populations and the potential for adaptive responses to warming by genetically diverse populations.

References

- Treml EA, Ford JR, Black KP, Swearer SE. Identifying the key biophysical drivers, connectivity outcomes, and metapopulation consequences of larval dispersal in the sea. *Mov Ecol.* 2015 Jul 15;3(1):17. doi: 10.1186/s40462-015-0045-6. PMID: 26180636; PMCID: PMC4502943.
- Hackerott, S., Martell, H. A., & Eirin-Lopez, J. M. (2021). Coral environmental memory: causes, mechanisms, and consequences for future reefs. *Trends in Ecology & Evolution*, 36(11), 1011-1023.
- Ogle, K. et al. Quantifying ecological memory in plant and ecosystem processes. *Ecol. Lett.* 18, 221–235 (2015).
- Peterson, G. D. Contagious disturbance, ecological memory, and the emergence of landscape pattern. *Ecosystems* 5, 329–338 (2002).
- Thomas, Luke, Elora H. López, Megan K. Morikawa, and Stephen R. Palumbi. "Transcriptomic resilience, symbiont shuffling, and vulnerability to recurrent bleaching in reef-building corals." *Molecular ecology* 28, no. 14 (2019): 3371-3382.
- Dziedzic, K. E., Elder, H., Tavalire, H., & Meyer, E. (2019). Heritable variation in bleaching responses and its functional genomic basis in reef-building corals (*Orbicella faveolata*). *Molecular Ecology*, 28(9), 2238-2253.
- Ainsworth, T. D., Heron, S. F., Ortiz, J. C., Mumby, P. J., Grech, A., Ogawa, D., ... & Leggat, W. (2016). Climate change disables coral bleaching protection on the Great Barrier Reef. *Science*, 352(6283), 338-342.

“Machine-Learning predictions of connectivity, biodiversity and resilience in the Coral Triangle”

L. Novi and A. Bracco

Reviewer 1:

“This paper proposes a framework based on machine learning to overcome these problems and evaluate the concurrent role of physical connectivity and ENSO in shaping the biodiversity of the CT and surrounding areas. Their analysis is based on sea surface temperature anomalies and unsupervised network analysis and allows for assessing both connectivity and bleaching recovery potential of reef ecosystems over unprecedented large space and time scales with uniform coverage. I think this work is interesting, innovative, and well-written. Overall, I have no major concerns about this paper. Yet there are some minor issues that need to be addressed before being accepted.”

Answer to Reviewer 1:

- *“Line 69: ‘ δ -MAPS’, what does this abbreviation refer to? This term was not defined earlier.”*

We acknowledge a definition of δ -MAPS at that point of the manuscript was still missing. We have thus updated it as follows: “ δ -MAPS, a complex-network algorithm developed for dimensionality reduction and network inference and related to clustering, multivariate statistics and community detection”. We thank the referee for having noticed. It is the name of the ML tool, not an acronym.

- *“Fig.1: Both the units for latitude and longitude are missing.”*

We have corrected the typo, and included the units for longitude and latitude in Figure 1. They are both [°]. We thank the referee for noticing.

- *“Line 79: I don’t think everybody knows what ‘UNEP-WCMC’, ‘WRI’, ‘TNC’, ‘USGS’ represent. Full names are needed.”*

We have now included the full names in the Caption. We have additionally included a reference to the data DOI for the Global Distribution of Coral reefs database.

- *“Line 92-114: The first two paragraphs in the results section were more about the methods rather than the results. I would suggest moving much text to the methods while keeping a brief description of the main approach used to generate the results.”*

We followed the reviewer suggestion and moved most of the details in the “Materials and Methods” section, renamed “ δ -MAPS : ecoregions and connectivity inference through SSTa”, while keeping in “Results” only what we believe is strictly necessary to understand the subsequent outcomes.

- “ Line 132-133: ‘in agreement with(32)’ . Line 163-164: ‘first introduced by(35)’ Line 182: ‘implementation in(37)’ Line 197: ‘see for example Fig. 3 in(3)’ Line 218: ‘38, and 39,40 for an overview of the product’ Line 244: ‘as conservation priorities by(43)’ .
Such a way of citation seems a bit strange. When I read these sentences, they look unfinished, as the numbers do not really belong to those sentences, but rather link to the references.”

We re-arranged the above sentences as suggested, by completing them with a clearer linkage to the cited works. For example ‘in agreement with (32)’ was turned into ‘in agreement with previous outcomes from Treml and colleagues (32)’ (and similar improvements are applied to the other indicated sentences as well).

Reviewer 2

“The manuscript brings an interest framework to assess connectivity at large scale and derive a spatial-prioritisation mechanism with potential to inform decision-making. The study derives from previously published work by the authors and looks at Sea Surface Temperature Anomalies (SSTa) over time and space to identify spatial domains or regions that likely to be highly correlated temporal activity. Based on derived correlations between SSTa and currents, the authors used 60 days as the Pelagic Larvae Duration, or time the coral larvae spends in the water column, to identify the connectivity among reefs and then perform the clustering of regions across the time series, discretised by ENSO phases. Using the inferred domains or regions, the authors expanded the work in two ways. First, by exploring the hypothesis that ENSO phases drive biodiversity in the Coral Triangle by exploring the contribution of connectivity using a centrality analysis. Second, the authors infer the recovery

of reefs to bleaching events based on the exposure to cumulative thermal stress predictions (Bleaching Alert from NOAA) and connectivity levels.

Overall, the manuscript is well-written, presented in a logical format to facilitate the interpretation despite the complexity of the topic and the approach is well-supported by previous research. This work presents a valuable contribution in terms of spatial and temporal variability in the definition of bioregions and connectedness across reef systems. Well done!

However, the following points need to be considered and addressed in this work before it follows its natural course to publication:”

Answer to Reviewer 2:

- “Assumptions and weaknesses in the approach need to be clearly stated and discussed. While the study is supported by body of research, this work is highly correlative in nature and make use of several assumptions that should be clearly discussed to outline further work and potential weaknesses. For example, I would appreciate a discussion considering, but not limited to, the following points:

- The effect of non-anomalous years, in terms of SST, on the patterns of connectivity. The study bases the regional clustering on SST anomalies. Would there be any difference in the

spatial patterns if considering connectivity during periods when SST is within its climatological range?"

The ecoregionalization and connectivity procedure is based on SST anomalies, i.e. detrended and deseasonalized SST, which are linked by construction to mesoscale variability in the ocean, which may differ, because mean conditions differ, in El Nino, La Nina or neutral years. The idea is the following: currents and mesoscale variability at scales of 30-300 km determine ocean advection. By doing so they influence larval transport *and* SST transport at these scales. The impact on SSTs determine the time evolution of the anomalies at these scales, which has nothing to do with the year being anomalous. SST anomalies are modulated by flowing geostrophic currents (which also imprint the SSH anomalies signal). Indeed, SST anomalies (and only the anomalies) show a strong coupling with the layers underlying the ocean surface through horizontal oceanic currents. In other words, once organized in domains they provide information equivalent to the eddy kinetic energy field. The SST field anywhere has anomalies, which are always present as deviation from the mean because eddies and mesoscale circulation exist. In the tropical Pacific the mean (in SST but also kinetic energy and very obviously SSH) is significantly different among El Nino, La Nina and neutral years, and therefore we used the climatology in each 'group' to calculate the anomalies that inform us about the mesoscale advection. In the Coral Triangle this is a required step (again, because the climatology of El Nino year is very different from that of neutral or La Nina year). In other regions, the clustering may not be required.

We acknowledge that this point might not be as clearly stated in the methods as needed. So, we have slightly extended the "Materials and Methods" section with a related clarification before detailing the algorithm (initial ~10 lines of Materials and Methods in the revised version).

- "- The effect of larval mortality and settlement competency characteristics in defining population connectivity. Previous studies suggest marine population connectivity is primarily influenced by the duration of the pelagic larval stage (PLD, considered here), but also the time-bound mortality of larvae and the extent of the pre-competency window. I must acknowledge that such information is species specific, and the availability of such data is sparse and a simpler approach, such as the one used here is justified. However, it is important to acknowledge the limitations of this simpler approaches."

We thank the referee for this contribution, as we believe discussing this point will improve the rigor of our manuscript. In particular, we extended the Discussion section including what suggested, as follows: "The proposed approach brings about an effective and fast tool to explore a rather complex set of processes acting in concert and, as such, introduces important simplifications to the biological-physical interactions at play. For example, we neglect the species-specific nature of larval mortality and settlement competency characteristic and use only information about pelagic larval duration (PLD) (see Methods). The paucity of detailed data over the whole study area justifies the usage of the PDL alone, but for single-species studies may not be sufficient."

- "Centrality of networks do not equate biodiversity. While the assessment of centrality is strongly linked to the potential of maintaining biodiversity, this works appears to use centrality as an estimate of biodiversity. I loved the approach of using web-page network analyses to explore the contribution of connectivity across ENSO phases to identify the contribution from different bioregions. The discussion is very appropriate from the potential of view of temporal variability in the contribution of larvae across different bioregions to potential generate high biodiversity over geological periods, but the point of this analyses should be centred on the maintenance of such biodiversity. It is possible that I misunderstood this point, but if so, it is a point that needs clarification."

We thank the referee for this comment, as it provided room for improvements. We acknowledge that the CT biodiversity question is still under debate, and the proposed network metric provides a novel approach to biodiversity loss mitigation in the CT, rather than trying to solve the CT biodiversity origin conundrum with a single tool. We therefore modify the section “Biodiversity patterns: ENSO as driver of the CT biodiversity” (Results) to reframe the meaning of a high Biodiversity Score, now associated to a high level of “connectivity-modulated potential for biodiversity maintenance”. We also clearly stated that the “centrality” concept is linked to the potential maintenance of high biodiversity in the CT, rather than using it as a proxy for biodiversity in the area.

- “Also, the authors refer to two important hypotheses, center of accumulation and center of overlap, but does not expand on them. This point needs more clarity.”

We acknowledge that clearer definitions of these hypotheses are needed. Thus, we extend this point when the hypotheses first appear in the text (in the Introduction). In particular we elaborate as follows: “It has been hypothesized that the CT is a center of origin: speciation occurred within the CT from geological times, with biodiversity gradients due to the ocean currents that limit expansion outwards. It was then argued that the CT owns its biodiversity to being a center of species accumulation. Finally, it was theorized that the CT is a center of overlap, where the geological separation of the Indian and Pacific Oceans, allowed the formation of different fauna through vicariance, followed by Indian and Pacific sister species overlap through species-ranges expansion, thus augmenting biodiversity.”

- “Recovery potential based on exposure to bleaching warnings. It is well-agreed in the literature that reefs that are highly connected are more likely to recover, and the authors make a good case of utilising this point to expand on this point to explore a mechanism to prioritise reefs based on their recovery potential. However, the authors also use Bleaching Alert Area datasets to explore time cumulative bleaching stress and argue that the recovery to bleaching depends on the previous number of bleaching occurrences, as well as connectivity. To this point, the authors suggest that low levels of previous bleaching events is an indicative of rapid recovery, while also considering connectivity. While there is strong support for the latter point (i.e., connectivity support recovery), I have concerns about the use of Bleaching Alert Area and the use of bleaching exposure to explore recovery potential for the following reasons:

- What justifies the assumption that low cumulative exposure to previous bleaching events equates to rapid recovery to a bleaching event? A body of work in the literature has documented the effect of ‘stress hardening’, suggesting various mechanisms where coral acquire and maintain stress tolerance through past environmental stress (see references below). Perhaps, it could be argued that low exposure will mean that coral cover is higher and community composition is more stable in the ecoregion, therefore justifying a higher supply of larvae that augments to the point around connectivity. However, based on the statement around stress-hardening, these communities could also be more vulnerable, or less resistant, to future bleaching events, which could affect the rate of recovery.

We thank the referee for this contribution. Indeed, we updated the “Discussion” section to clearly state the limitations associated with the assumption that a low-exposed reef will on average experience a more rapid recovery. We also speculate that the procedure used to evaluate CMBV and RPS, could be analogously applied to future scenarios (by feeding our algorithm with SSTa from numerical models under future scenarios), rather than using the current RPS as a forecasting tool for more than a few years.

- “If previous exposure to bleaching is justified, why using the Bleaching Alert Area instead of actual metrics of thermal stress (e.g., SSTA and DHW)? Bleaching Alert Area from NOAA Coral Watch is a forecasting tool based on Degree Heating Week thresholds over a moving window of 7 days to provide a warning system for potential bleaching events. Alert 1 and 2 values are based on SST levels above documented thresholds for thermal stress and probability of bleaching and mortality (DHW >4 and >8, respectively). However, for hindcasting historical bleaching events, DHW and SSTA would be more appropriate. Perhaps the previous analysis on network centrality would be more appropriate here as a proxy for maintenance of genetic diversity across populations and the potential for adaptive responses to warming by genetically diverse populations.”

We thank the reviewer for this comment, as it provides rooms for further robustness assessment. The choice of using monthly composites of 7-day maximum composites of b.a.a from NOAA CRW resides in the fact that it outlines the areas where coral bleaching heat stress reaches various levels, based on (NOAA’s) satellite sea surface temperature (SST) monitoring, including the heat stress condition determined by the seven most recent daily value pairs of CRW HotSpot (related to SSTA) daily values and Degree Heating Week daily values, i.e. accounting for and including at once both these metrics (DHW and HotSpots). The HotSpot values measure the occurrence and magnitude of instantaneous heat stress, potentially resulting in coral bleaching over a scale range between 0 and 5°C (with <1°C for bleaching). The DHW, as stated by the referee, refers to accumulated heat stress leading to bleaching and potentially death of corals, with bleaching occurring for DHW reaching 4^o-week and widespread bleaching and significant expected mortality when DHW achieves 8^oweek. It is not expected to provide different outcomes and indeed we repeated the analysis using the monthly composite of max. DHW from NOAA CRW (DHW in the following). We made the following choice of including three potential “classes” of risk:

for $0 < \text{DHW} < 4$: “no stress” to “warning” for possible bleaching. We assign a value of 2.

for $\leq 4 \text{ DHW} < 8$: significant bleaching expected. We assign a value of 3.

for $\text{DHW} \geq 8$: widespread bleaching and likely mortality. We assign a value of 4.

We assign 0 elsewhere. Then, we sum these DHW classes values over time in each ENSO phase (t.s.dhw), and we compute the Connectivity Modulated Bleaching Vulnerability $\text{CMBV} = \text{t.s.dhw}/\sigma^+$. We proceed to the RPS calculation as usual, using this DHW-based CMBV (instead of t.s.baa-based). The resulting RPS are nearly identical (as they should be) to those obtained with t.s.baa (compare new Suppl Fig. S13, with Fig. 5 and Suppl Fig. S11, S12), thus the procedure is robust to the selection of bleaching metrics among those tested (DHW and b.a.a.). We included this finding in the manuscript (Results section and Suppl Fig S13).

The following suggested references have also been added where appropriate.

Treml EA, Ford JR, Black KP, Swearer SE. Identifying the key biophysical drivers, connectivity outcomes, and metapopulation consequences of larval dispersal in the sea. *Mov Ecol.* 2015 Jul 15;3(1):17. doi: 10.1186/s40462-015-0045-6. PMID: 26180636; PMCID: PMC4502943.

Hackerott, S., Martell, H. A., & Eirin-Lopez, J. M. (2021). Coral environmental memory: causes, mechanisms, and consequences for future reefs. *Trends in Ecology & Evolution*, 36(11),

1011-1023.

Ogle, K. et al. Quantifying ecological memory in plant and ecosystem processes. *Ecol. Lett.* 18, 221–235 (2015).

Peterson, G. D. Contagious disturbance, ecological memory, and the emergence of landscape pattern. *Ecosystems* 5, 329–338 (2002).

Thomas, Luke, Elora H. López, Megan K. Morikawa, and Stephen R. Palumbi. "Transcriptomic resilience, symbiont shuffling, and vulnerability to recurrent bleaching in reef-building corals." *Molecular ecology* 28, no. 14 (2019): 3371-3382.

Dziedzic, K. E., Elder, H., Tavalire, H., & Meyer, E. (2019). Heritable variation in bleaching responses and its functional genomic basis in reef-building corals (*Orbicella faveolata*). *Molecular Ecology*, 28(9), 2238-2253.

Ainsworth, T. D., Heron, S. F., Ortiz, J. C., Mumby, P. J., Grech, A., Ogawa, D., ... & Leggat, W. (2016). Climate change disables coral bleaching protection on the Great Barrier Reef. *Science*, 352(6283), 338-342.

REVIEWERS' COMMENTS:

Reviewer #1 (Remarks to the Author):

The authors have addressed the comments quite well. I have no more suggestions.